# Influence of Material Composition on Structure, Surface Properties and Biological Activity of Nanocrystalline Coatings Based on Cu and Ti

**Damian Wojcieszak [1,\*], Malgorzata Osekowska [2], Danuta Kaczmarek [1], Bogumila Szponar [2], Michal Mazur [1], Piotr Mazur [3] and Agata Obstarczyk [1]**

[1] Faculty of Microsystem Electronics and Photonics, Wroclaw University of Science and Technology, Janiszewskiego 11/17, 50-372 Wroclaw, Poland; danuta.kaczmarek@pwr.edu.pl (D.K.); michal.mazur@pwr.edu.pl (M.M.); agata.obstarczyk@pwr.edu.pl (A.O.)

[2] Institute of Immunology and Experimental Therapy, Polish Academy of Sciences, Weigla 12, 53-114 Wroclaw, Poland; osekowska.malgorzata@gmail.com (M.O.); bogumila.szponar@hirszfeld.pl (B.S.)

[3] Institute of Experimental Physics, University of Wroclaw, Max Born 9, 50-204 Wroclaw, Poland; pioma@ifd.uni.wroc.pl

\* Correspondence: damian.wojcieszak@pwr.edu.pl

**Abstract:** In this paper, the influence of material composition on structure and surface properties of bioactive coatings based on Cu and Ti is described. Nanocrystalline coatings were prepared by innovative pulsed DC magnetron sputtering. For their preparation, a multi-magnetron system was used in order to obtain films with various copper content. The main goal of our work was the complex analysis of biological activity of Cu-Ti films in comparison with their material composition and surface state. Antimicrobial activity (for *E. coli* and *S. aureus*), as well as the impact on cell viability (L929 line), were investigated. The physicochemical properties were examined with the aid of X-ray diffraction, scanning electron microscopy, X-ray photoelectron spectroscopy, and atomic absorption spectroscopy. It was found that all prepared films were nanocrystalline and bactericidal, but their cytotoxicity was related to the Cu-content in the film. Complex analysis of the bioactivity was developed in relation to the copper ion migration process. Moreover, manufacturing of antibacterial films with stimulating action on L929 cell line was possible.

**Keywords:** antibacterial; cytotoxic; functional coating; Cu-Ti thin film; sputtering

## 1. Introduction

Many different materials are used in medical applications. Especially important are metals and their alloys [1–4], polymers [5,6], or various types of ceramics (e.g., hydroxyapatite) [4,7–13]. Attention should also be paid to thin-film coatings [14–16]. Titanium is very important in this area of application. Ti is widely used as a biomaterial for implants or cardiovascular devices due to its biocompatibility, high osseointegration capacity, and high specific strength [17–23]. The most important feature of Ti is high corrosion resistance [17,20,24]. The native oxide layer formed on its surface also favors the adhesion with bone tissue. The widespread use of titanium is also related to its low elastic modulus (55 to 85 GPa), which is similar to cortical bone tissue (from 20 to 30 GPa) [25]. Despite those advantages, titanium has several limitations. The most important disadvantage (and, simultaneously, also the advantage) is its bio-neutrality. This feature can be overcome by doping with bioactive metals, i.e., copper [23]. Cu represents one of the most promising metals for biomedical applications [26]. It is a ductile metal with a relatively low corrosion resistance. Forming alloys with other metals allows very wide application of Cu [27]. The deliberate use of copper for medical purposes took place

at the beginning of the 20th century [28,29]. Currently, the antimicrobial properties of copper are being rediscovered [26]. Cu exhibits strong cytotoxicity even when compared with such bioactive elements as Al, Ag, V, Mn, Cr, Zr, Nb, or Mo [30]. Its effectiveness was confirmed in the inactivation of bacteria [31–35], viruses [36–38], algae [39], parasites [34], and fungal pathogens [40–43]. Copper participates in the synthesis of about 30 proteins and is a cofactor of many key enzymes; therefore, its deficiency in the body is dangerous for health and life [44,45]. Cu is essential for the proper organism growth and a development of tissue structures [46,47]. It plays an important role in all stages of wound healing (in homeostasis, inflammation, proliferation, and remodeling) [48–50]. Misbalancing the distribution of copper is dangerous for the cell, and its excess, as well as deficit, is unfavorable. In the case of copper excess, a cytotoxic effect may occur. There are many potential mechanisms of Cu-toxic effect on a cell that can interact with each other. Toxicity of copper is mainly related to overproduction of (reactive oxygen species) ROS, which triggers a cascade of biochemical and physiological changes at the cellular level [51,52]. Other mechanisms are associated with the location and a number of its transporters in the cell and reduced lipid metabolism [53]. High activity of copper in contact with microbes or cells is related to the migration of its ions to the surrounding environment. However, copper is an essential trace element found in every cell, and its highest concentration occurs in the tissues of organs with the highest metabolic activity (including liver, brain, etc.) [54]. The well-regulated distribution process is responsible for its proper level in the body. In the body of an adult human with a mass of approximately 70 kg, copper is present in the amount of 75–100 mg. The daily supply of copper in the diet depends on the sex, age, and health of the person. An adult should not consume more than 1.35 mg per day [55]. The defense against copper toxicity is associated with a decrease in absorption of this element from the digestive tract or its greater excretion [56]. Copper in the body occurs at two oxidation states ($Cu^{2+}$ and $Cu^{1+}$). As it was reported in previous works [30,57,58], Cu(I) ions are the most active form of copper. In order to get into the cell, $Cu^{2+}$ ion must be reduced. It is worth nothing that the CTR1 membrane protein, having an affinity for $Cu^{1+}$, transports about 80% of copper in the body without energy expenditure [59].

As it was mentioned, the properties of titanium and copper alone have been well understood [60–63], but the combination of them allows manufacturing of a completely new materials with unique properties. According to previous works [64,65], the main advantage of such a material can be a specific, selective bioactivity. For this reason, the main aim of this work was a complex analysis of the bactericidal activity and the cytotoxicity of nanocrystalline Cu-Ti coatings for application in medicine, architecture, innovative electronic devices, etc. The first work about the bioactivity of Cu-Ti alloys was published by Holden et al. in the 1950s [66]. However, even at present, the knowledge about this material is still limited [30]. Addition of Cu improves mechanical properties of Ti [67], as well as increases hardness and tensile strength [65]. Moreover, it increases susceptibility to grinding, which is important in implant technology [28]. Release of such strong bactericidal agents as copper ions can neutralize all bacteria on the implant surface, even when they form a resistant biofilm [68–70]. Properties of Cu-Ti coatings were briefly reported by Stranak et al. [71] and mentioned in our previous work [72], but a complex analysis has not yet been described. Moreover, its effect on both bacteria and eukaryotic cells was neglected. Therefore, in this work, a description of manufacturing and properties of a new-type multifunctional Cu-Ti coatings was presented. The bioactivity was analyzed in relation to copper and titanium content in the film and structural properties, as well as the ion migration process and the copper oxidation level. Based on the obtained results, a model of film interaction was proposed.

## 2. Materials and Methods

### 2.1. Thin Films Manufacturing

Thin films were prepared by pulsed DC magnetron sputtering [73–79]. Desired material composition was obtained by simultaneous sputtering (with the appropriate power) of several

targets made of Cu and Ti [78,80–86]. For all prepared films, the deposition processes were carried out in argon plasma at a pressure of $3 \times 10^{-2}$ mbar, which was obtained with an argon flow of approximately 6 mL/min. Sputtered materials had a form of a metallic titanium and copper targets (diameter: 30 mm, thickness: 3 mm, purity: 99.995%). They were simultaneously sputtered using one, two, or three magnetrons, supplied with adequate power. The difference in sputtering rate of both materials was taken into account. The distance between the target and the substrates ($SiO_2$ and Si) mounted on a rotary drum was 90 mm. The deposition time was equal to 20 min. Detailed data on the technological parameters of sputtering processes are collected in Table 1.

**Table 1.** Composition and deposition parameters by pulsed DC magnetron sputtering of thin films based on copper and titanium.

| Thin Film | Composition [at. %] | | $P_{Ar}$ [mbar] | Sputtering Power [W] | | | t [nm] |
|---|---|---|---|---|---|---|---|
| | Cu | Ti | | Magnetron 1-target Ti | Magnetron 2-target Cu | Magnetron 3-target Ti | |
| Cu | 100 | - | | - | 300 | - | 880 |
| $Cu_{83}Ti_{17}$ | 83 | 17 | | 150 | 200 | 150 | 790 |
| $Cu_{53}Ti_{47}$ | 53 | 47 | $3 \times 10^{-2}$ | 400 | 100 | 400 | 650 |
| $Cu_{25}Ti_{75}$ | 25 | 75 | | 400 | 50 | 400 | 540 |
| Ti | - | 100 | | 400 | - | 400 | 420 |

Designations: t–thickness, $P_{Ar}$–pressure of argon during sputtering.

## 2.2. Characterization of Physicochemical Properties

Structural properties were analyzed using X-Ray Diffraction (XRD, Pananalytical, Malvern, UK). The measurements were performed employing PANalytical Empyrean PIXel3D powder diffractometer with Cu K$\alpha$ X-ray ($\lambda$ = 1.54056 Å). Surface properties were determined by Scanning Electron Microscopy (SEM, Thermo Fisher Scientific, Waltham, MA, USA) and X-Ray Photoelectron Spectroscopy (XPS, Specs, Berlin, Germany). Observations of surface and cross-section morphology were made with the use of FEI Helios NanoLab 600i SEM coupled with an EDS spectrometer. Comparison of the intensity of $Cu_{L\alpha}$ and $Ti_{K\alpha}$ emission lines allowed us to determine the amount of each element. The surface chemical state was determined using a Specs Phoibos 100 MCD-5 hemispherical analyzer (Specs) and a Specs XR-50 X-ray source with Mg K$\alpha$ (1253.6 eV) beam. As a standard operation, all obtained XPS spectra were calibrated to the binding energy of adventitious C1s peak at 284.8 eV. Based on the XPS spectra, the percentage of signal from copper ions at +1 and +2 oxidation state was calculated. This estimation was made with an accuracy of ca. ±2%. Moreover, the thickness was determined with the aid of Talysurf CCI optical profilometer (Taylor Hobson, Leicester, UK). The atomic absorption spectrometry (AAS) method was used to study the amount of released ions. The measurements were made using a GBC AVANTA spectrometer (GBC Scientific Equipment, Braeside, Australia), calibrated in the so-called least squares concentration mode. The volume of taken samples was 25 mL and was acidified with nitric acid (0.25 mL of 65% acid per 25 mL of sample). The results were calculated taking into account the size of the samples. The accuracy of the estimation can be assumed at ca. ±2%.

## 2.3. Measurements of Antimicrobial Activity of Thin Films

To determine the level of antimicrobial activity of nanocrystalline films, standard methods were adapted. Selected bacterial strains (in suspension) were directly exposed by placing on a solid material. The subject of the research were as-deposited thin films on $SiO_2$ substrates with an area of 30 cm$^2$ (in samples with dimensions of $1 \times 1$ cm$^2$). Determination of their activity was performed by exposure to bacterial suspension and examination of the bactericidal effect by means of a modified dilution method. *Escherichia coli* PCM 1144 (Gram-negative) and *Staphylococcus aureus* PCM 260 (Gram-positive) bacteria were used in the studies due to different cellular shields. Prior to the experiment, samples were sterilized under a UV lamp. A prepared suspension of bacteria (with optical density OD ~0.5)

was exposed to the test material in a 24-well sterile plate and incubated for 24 h at 37 ± 1 °C. The inoculum was collected at time intervals: 0, 2, 4, 6, and 24 h of the experiment. The dilution method was used to determine the amount of live bacteria after contact with the film. A geometric dilution series was made, and then 200 μL was harvested and plated on agar plates, incubated for 24 h at 37 °C, and colonies were counted. The result was given in colony forming units (CFU) per mL. The results are the average value of two independent measurements series (the difference between them was less than 2–3%).

*2.4. Studies of Thin Films Cytotoxicity*

Cytotoxicity was investigated using the L929 murine fibroblast cell line (from ATCC). The cellular response for direct contact of cells with the material, as well as indirect contact with the extracts from the materials, was tested. All procedures were performed under aseptic conditions. The level of cytotoxicity was determined based on the MTT and clonogenic test.

Cells were cultured under constant conditions (5% $CO_2$, 37 °C, humidity >95%) in a SteriCycle 381 Thermo Scientific (Waltham, MA, USA) incubator, stored in liquid nitrogen and in culture fluid (DMEM, Dulbecco's Modified Eagle Medium) with 25 mM HEPES and 4.5 g/L glucose (Lonza Sales Ltd., Basel, Switzerland, 20% FBS (Fetal Bovine Serum): Lonza Sales Ltd, 7% Deso: Sigma Aldrich, St. Louis, MO, USA). To stabilize their metabolism, two passages were performed using 0.25% Trypsin-ethylenediaminetetraacetic acid (EDTA) (Sigma Aldrich). A solution of trypsin and ethylenediaminetetraacetic acid (EDTA) made it possible to unstick the monolayer from the vessel and dissolve the collagen joints between the cells, which allowed us to obtain a suspension from individual cells. During cell cultures, a medium with DMEM composition with 25mM Hepes, 4.5g/L glucose (Lonza Sales Ltd.), together with L-glutamine, 10% fetal bovine serum FBS (Lonza Sales Ltd.) was used. In order to show an appropriate number on the culture vessels, counting of live cells was done using an automatic cell counter from Digital Bio (New York, NY, USA) and a haemocytometer. The accuracy of the estimation can be assumed at ca. ±2%.

2.4.1. Extracts from Tested Films

Extracts were prepared in accordance with PN-EN ISO 10993-12:2009 [87]. They were made under sterile conditions under the MSC Advantage 1.2 laminar chamber (Thermo Scientific, Biohazard, Waltham, MA, USA). Thirty centimeters squared of material on $SiO_2$ substrates coated with both sides in the form of squares (with dimensions of $10 \times 10$ mm$^2$) was used. After 20 min of sterilization with UV light, 10 mL of full-value culture medium was added to each series of samples. They were incubated in solid, sterile culture conditions for 24 to 240 h (SteriCycle 381 Thermo Scientific, Waltham, MA, USA). After incubation, the material was removed from the tube, and the medium was considered as 100% extract. In each experiment, control samples were made, i.e., negative control, based on full-value culture medium and positive control with phenol (Merck KGaA, Darmstadt, Germany), at a concentration of 1.5 mg/cm$^3$ and 4mg/cm$^3$.

2.4.2. MTT Test

Evaluation of the metabolic activity of L929 cells in a contact with tested materials was performed in accordance with the PN-EN ISO 10993-5:2009 procedure [88]. During the experiment, mouse fibroblasts were seeded at a density of $1 \times 10^5$/mL per well in a 96-well NUNC plate. After 24 h of incubation in a nutrient medium, when the cell surface coverage was about 60%, the culture medium was changed into prepared extracts at the following concentrations: 100%, 50%, 25%, and 12.5%. After incubation, microscopic documentation of cell morphology was made and rinsed with 100 μL of buffered saline (PBS). A solution of MTT (5 mg of 3-[4,5-dimethylthiazol-2-yl]-2,5-diphenyl bromide dissolved in 1 mL of PBS and 9 mL of the culture medium) was added to each well and incubated under constant conditions (37 °C, 5% $CO_2$, humidity >95%). After a 2-h incubation, solubilization was performed by adding 100 μL of isopropyl acid alcohol to each well (20 mL of isopropyl alcohol and

76 μL of 36% hydrochloric acid). This solution was stirred and incubated at room temperature for 30 min. The degree of metabolic activity of selected cell line was determined on the basis of absorbance measurements at λ = 570 nm (Epoch, Biotek Spectrophotometer, Winooski, VT, USA) [88]. The results are the average value of two independent measurements series. The accuracy of the estimation was in range below 2%.

### 2.4.3. Clonogenic Test

To determine the ability of L929 mouse fibroblasts to proliferate after contact with extracts from the tested materials, a clonogenic test was used. As a part of the experiment, 100 mouse fibroblast cells were plated in each well on sterile six-well plates (NUNC Companies, Roskilde, Denmark). After 8 days of proliferation under constant culture conditions (37 °C, 5% $CO_2$, humidity >95%), cultures were fixed with 96% ethyl alcohol (3 mL/well/5 min) and stained by May Grunwald and Giemsa. A 5% Giemsa dye solution prepared in a Sorensen buffer (2mL/well/5 min) was used for staining. The counting of the colonies was performed using a CKX 41 Olympus contra-phase microscope (Olympus, Shinjuku, Tokio, Japan). The results are the average value of two independent measurements series. The accuracy of the estimation can be assumed at ca. ±2%.

## 3. Results

Material composition of prepared coatings was determined based on the results of X-ray microanalysis. In Figure 1, recorded EDS spectra are shown. It was found that copper content in Cu-Ti films was 83 at.%, 53 at.%, and 25 at.%. In Table 1, composition and sputtering parameters of coatings are presented. It also contains the results of thickness measurements, which were in the range from 420 nm to 880 nm.

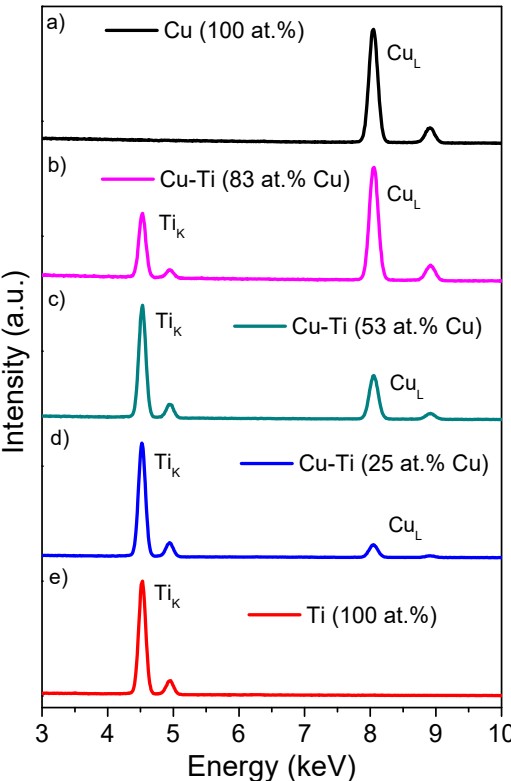

**Figure 1.** EDS spectra of: (**a**) Cu, (**b**) $Cu_{83}Ti_{17}$, (**c**) $Cu_{53}Ti_{47}$, (**d**) $Cu_{25}Ti_{75}$, and (**e**) Ti thin films prepared by magnetron sputtering.

In Figure 2, SEM images of surface topography and thin film cross-section are shown. As it can be observed, all coatings had high homogeneity. In the case of Cu film, its columnar structure can be seen (Figure 2a$_1$). The diameter of the columns was about 200 nm. In addition, there were empty spaces (voids) with the width of 30 nm between neighboring columns. Seventeen at.% addition of Ti resulted in a film ($Cu_{83}Ti_{17}$) with a much more densely packed structure (Figure 2b$_1$). The diameter of the columns was about 50 nm, and a lack of voids between them was observed. Increase of titanium content in Cu-Ti film up to 47 at.% caused a significant reduction in the diameter of the columns (20–30 nm), but in the film structure, inter-column spaces appeared with dimensions reaching up to approximately 10 nm (Figure 2c$_1$). Further increase of titanium in the film (75 at.%) caused the change of microstructure character from columnar to granular. As it can be seen in Figure 2d$_1$, the $Cu_{25}Ti_{75}$ coating consisted of long grains with a length not exceeding approximately 100 nm. Lack of empty spaces between densely packed grains was also observed. The Ti coating had a grain structure (Figure 2e) with grains up to 200 nm. Moreover, the film composition had a significant impact on the surface topography. The roughness decreased proportionally to the amount of copper in their composition. Above 53 at.% of Cu, a significant expansion of surface texture (especially for a pure Ti film) can be seen.

More accurate information about the impact of the material composition on the structure of Cu-Ti thin films was obtained by X-ray diffraction. In the case of one-component Cu film (Figure 3a), reflexes from four crystallographic planes at the XRD-pattern were observed. They indicate the occurrence of the Cu phase. The average size of crystallites (*D*) was in the range from 12.4 nm to 45.4 nm, but the largest quantity was constituted by crystallites in size of 29.3 nm (Table 2). XRD results have shown that the $Cu_{83}Ti_{17}$ thin film was also nanocrystalline. Two separate phases were identified in its structure (Cu and $Cu_4Ti_3$). The top-phase was Cu, and it was built of crystallites with an average size of 9.9 nm. In turn, the $Cu_4Ti_3$ phase was composed of much smaller crystallites (in size of 4.8 nm), and its percentage share was lower. Reduction of copper content in the Cu-Ti film (down to 53 at.%) caused the Cu phase, consisting mainly of crystallites with *D* = 15.3 nm, to be dominant. Besides, presence of $Cu_4Ti_3$ phase (*D* = 2.6 nm) was also identified, but its amount was very small. In the case of the $Cu_{25}Ti_{75}$ film, XRD studies also revealed presence of Cu and $Cu_4Ti_3$ phases. The average size of crystallites of the Cu phase (predominant in its structure) was in the range from 10.2 nm to 24.9 nm. The $Cu_4Ti_3$ phase (less crystallized) was composed of crystallites with *D* = 4.8 nm. The last, one-component Ti thin film was also nanocrystalline. It was built of crystallites with an average size of 10.5 nm ÷ 16.6 nm. The largest quantity constituted crystallites with *D* = 16.6 nm.

The surface oxidation plays an important role in the film contact with living organisms. As shown in previous works [70,89], $Cu^{1+}$ ions have a much larger impact on biological structures than $Cu^{2+}$. Besides, a large amount of titanium ions at +4 oxidation state may suggest that the surface of the metallic film has undergone passivation, which has a positive effect on its stability and corrosion resistance. Therefore, the degree of surface oxidation was determined by photoelectron spectroscopy. Figure 4 shows the XPS spectra for the Cu2p and Ti2p states. The analysis showed that, for all films containing copper in the spectra for the Cu2p state, the Cu2p$_{1/2}$ and Cu2p$_{3/2}$ peaks were present at 952.3 eV and 932.4 eV, respectively. This indicates the presence of copper ions at 0 and +1 oxidation state. In addition, in the spectra for the Cu2p state, the peak at 934.6 eV can be seen. It testifies the presence of $Cu^{2+}$ ions in all nanocrystalline films. Detailed analysis of peak intensities allowed on calculation of percentage quantity of Cu(0) and Cu(I) ions in relation to Cu(II) ions. In the case of nanocrystalline Cu, $Cu_{83}Ti_{17}$, and $Cu_{53}Ti_{47}$ films, the quantity of $Cu^{0,1+}$ ions in the near-surface area was more than twice that of $Cu^{2+}$ ions. For the $Cu_{25}Ti_{75}$ film, the amount of $Cu^{2+}$ ions decreased to 12.3%. For coatings containing titanium in their composition, analysis of XPS spectra (Figure 4) showed that the difference in binding energy between the positions of Ti2p$_{1/2}$ and Ti2p$_{3/2}$ peaks was about 5.7 eV. This shows that titanium was at +4 state. In the spectra for the Ti2p state, peaks indicating on the presence of $Ti^{3+}$ ions (at 456 eV) can also be observed, but their percentage quantity was small (reaching only a few percent).

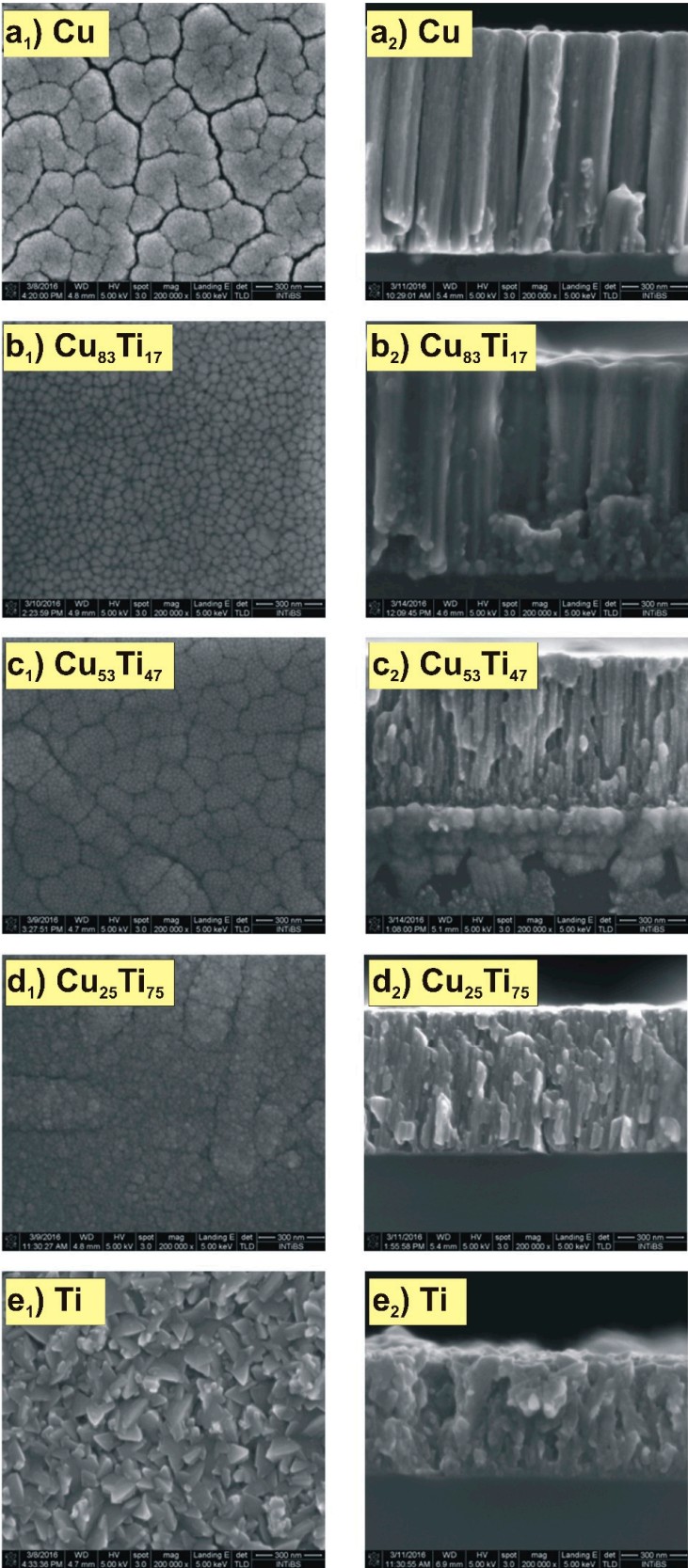

**Figure 2.** SEM images of prepared coatings: (**a**) Cu, (**b**) $Cu_{83}Ti_{17}$, (**c**) $Cu_{53}Ti_{47}$, (**d**) $Cu_{25}Ti_{75}$, and (**e**) Ti, as-deposited on Si substrates. Designations: (1) surface, and (2) cross-section.

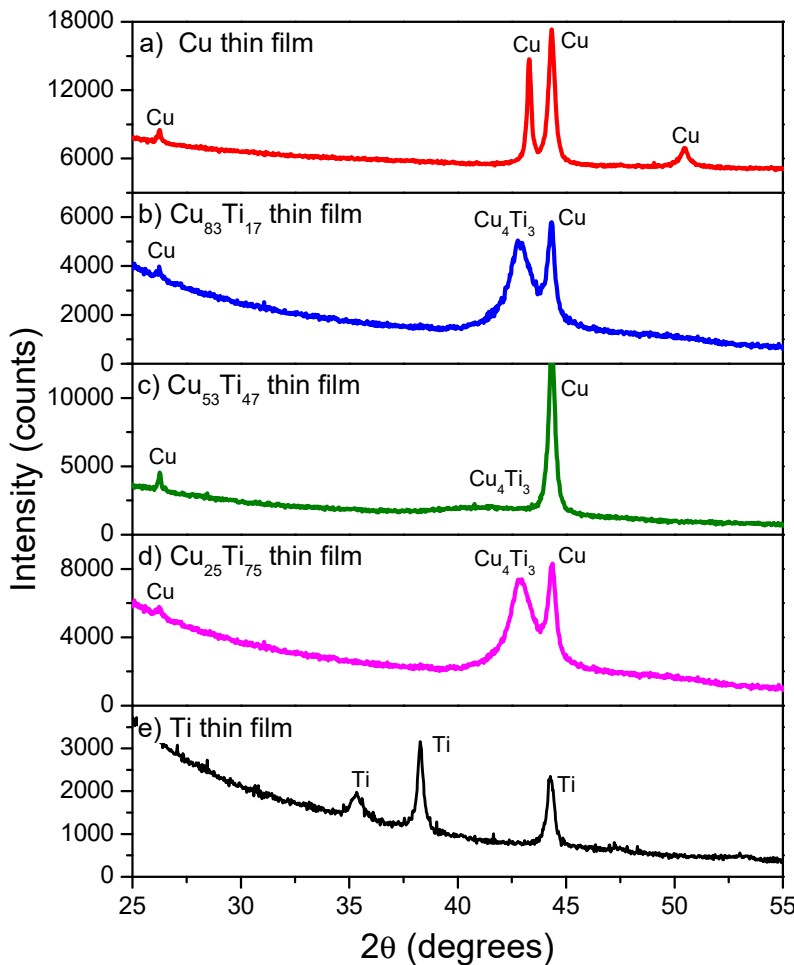

**Figure 3.** XRD patterns of prepared coatings: (**a**) Cu, (**b**) $Cu_{83}Ti_{17}$, (**c**) $Cu_{53}Ti_{47}$, (**d**) $Cu_{25}Ti_{75}$, and (**e**) Ti.

**Table 2.** Structural parameters of thin films based on Cu and Ti, determined by XRD.

| Thin Film | 2θ (°) | Phase | *d* (nm) | *D* (nm) |
|---|---|---|---|---|
| **Cu** | 26.42 | Cu | 0.2203 | 45.4 |
| | 43.31 | Cu | 0.1355 | 29.3 |
| | 44.33 | Cu | 0.1325 | 18.4 |
| | 50.44 | Cu | 0.1173 | 12.4 |
| **$Cu_{83}Ti_{17}$** | 26.25 | Cu | 0.2202 | 32.7 |
| | 42.66 | $Cu_4Ti_3$ | 0.1374 | 4.8 |
| | 44.32 | Cu | 0.1326 | 9.9 |
| **$Cu_{53}Ti_{47}$** | 26.24 | Cu | 0.2203 | 36.6 |
| | 42.36 | $Cu_4Ti_3$ | 0.1384 | 2.6 |
| | 44.34 | Cu | 0.1325 | 15.3 |
| **$Cu_{25}Ti_{75}$** | 26.24 | Cu | 0.2203 | 24.9 |
| | 42.70 | $Cu_4Ti_3$ | 0.1373 | 4.8 |
| | 44.41 | Cu | 0.1323 | 10.2 |
| **Ti** | 35.35 | Ti | 0.1647 | 10.5 |
| | 38.27 | Ti | 0.1525 | 16.6 |
| | 44.21 | Ti | 0.1327 | 14.9 |

Designations: θ–diffraction angle, *d*–interplanar distance, *D*–average crystallites size.

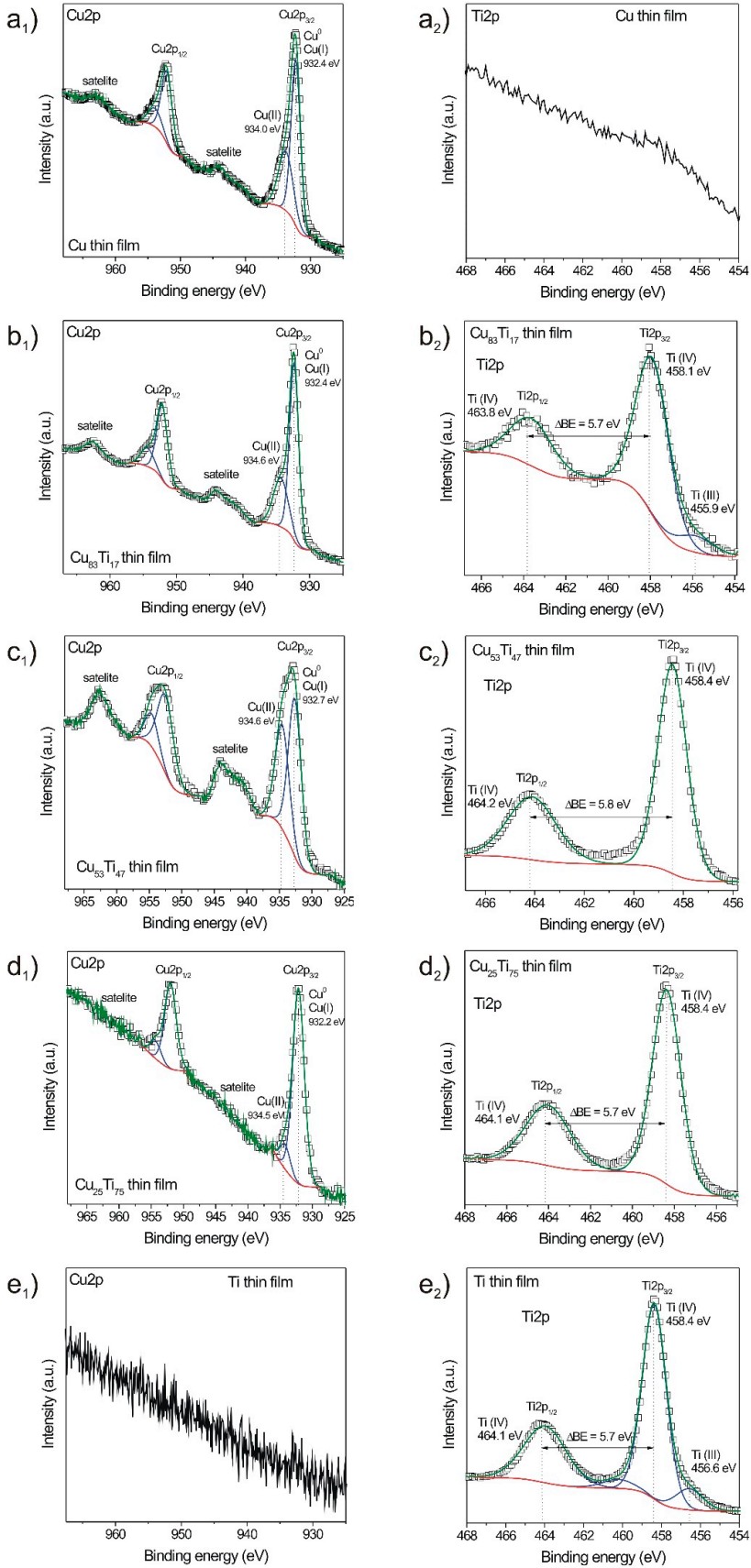

**Figure 4.** XPS spectra of nanocrystalline coatings: (**a**) Cu, (**b**) $Cu_{83}Tu_{17}$, (**c**) $Cu_{53}Ti_{47}$, (**d**) $Cu_{25}Ti_{75}$, and (**e**) Ti. Designations: (1) Cu2p state, and (2) Ti2p state.

Figure 5 shows images of agar media on which suspensions with *E. coli* and *S. aureus* bacteria were grown. They were exposed to the direct contact with the coating's surface. In the case of Cu, thin film inactivation of all bacteria of both strains was observed just after 2 h (Figure 5a). This means that the nanocrystalline Cu coating was characterized by a very strong antimicrobial effect. In turn, the Ti coating did not inhibit bacterial growth (Figure 5e). There was no significant reduction in the amount of microorganisms in the suspension. In Figure 5b, results of bactericidal tests of $Cu_{83}Ti_{17}$ film are shown. Similar to Cu film, after 2 h, all bacteria were neutralized. $Cu_{53}Ti_{47}$ thin film also exhibited high anti-microbial activity against both bacterial strains (Figure 5c, Table 3). Total neutralization occurred after 4–6 h. Further reduction of copper content in the Cu-Ti film to 25 at.% also had an impact on the level of antimicrobial activity (Figure 5d, Table 3). In the case of *E. coli*, antimicrobial activity was lower as compared to *S. aureus*.

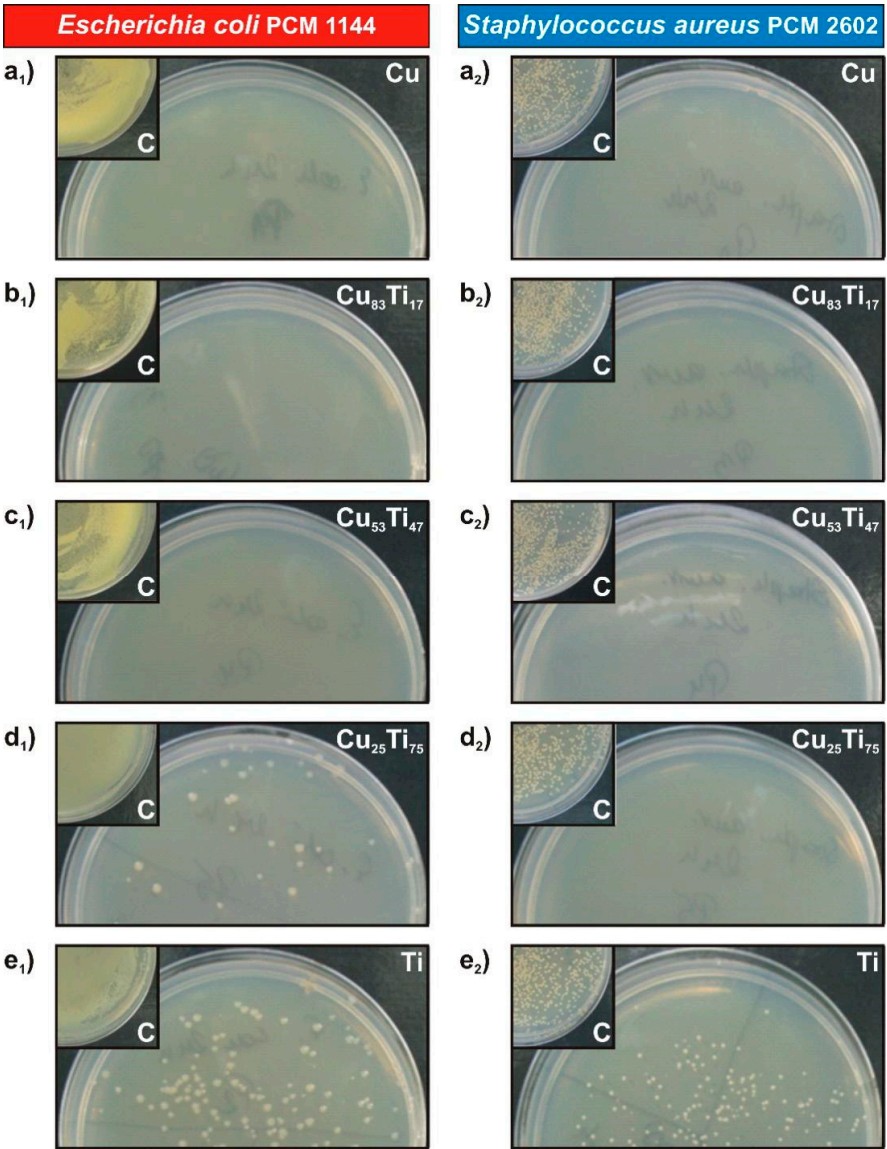

**Figure 5.** Antimicrobial activity of thin films based on Cu and Ti against (1) *E. coli* (PCM1141) and (2) *S. aureus* (PCM 2602), after 24 h exposure in contact with: (**a**) Cu, (**b**) $Cu_{83}Ti_{17}$, (**c**) $Cu_{53}Ti_{47}$, (**d**) $Cu_{25}Ti_{75}$, and (**e**) Ti. In the inset of each picture, a result for the control sample (incubation without contact with the film surface) is shown.

**Table 3.** Antimicrobial activity of thin films based on Cu and Ti for *E. coli* and *S. aureus*.

| Exposure Time (h) | Amount of Colony Forming Units per Milliliter (CFU/mL) | | | | |
|---|---|---|---|---|---|
| | Cu | $Cu_{83}Ti_{17}$ | $Cu_{53}Ti_{47}$ | $Cu_{25}Ti_{75}$ | Ti |
| | for *E. coli* | | | | |
| 0 | $7.3 \times 10^7$ | $7.3 \times 10^7$ | $7.3 \times 10^7$ | $7.3 \times 10^7$ | $7.3 \times 10^7$ |
| 2 | 0 | 0 | $1.0 \times 10^5$ | $7.2 \times 10^7$ | $5.1 \times 10^7$ |
| 4 | 0 | 0 | $5.0 \times 10^4$ | $7.0 \times 10^7$ | $6.0 \times 10^7$ |
| 6 | 0 | 0 | 0 | $5.4 \times 10^7$ | $5.0 \times 10^7$ |
| 24 | 0 | 0 | 0 | $1.3 \times 10^6$ | $9.2 \times 10^6$ |
| | for *S. aureus* | | | | |
| 0 | $6.2 \times 10^7$ | $6.2 \times 10^7$ | $6.2 \times 10^7$ | $6.2 \times 10^7$ | $6.2 \times 10^7$ |
| 2 | 0 | 0 | $5.0 \times 10^7$ | $2.0 \times 10^7$ | $3.7 \times 10^7$ |
| 4 | 0 | 0 | 0 | $1.3 \times 10^7$ | $2,1 \times 10^7$ |
| 6 | 0 | 0 | 0 | $4.8 \times 10^6$ | $7.0 \times 10^6$ |
| 24 | 0 | 0 | 0 | 0 | $6.0 \times 10^6$ |

The results of the study of morphological changes in L929 cells in contact with extracts from the examined films are shown in Figure 6. It was found that there is a close relation between the amount of copper in the coating and cell morphology, as well as with their confluence. In the case of Cu coating, the death of all L929 cells was found. Cell morphology was significantly different from these in the control culture. Fibroblasts with an interrupted cell membrane can be observed. It can be noticed that some of the adherent L929 cells were detached from the substrate, what also indicates the cytotoxic effect. In the case of $Cu_{83}Ti_{17}$ film, the level of toxicity was smaller (Figure 6). Only after 72 h was it comparable to the one noticed for Cu film. These cells were mostly peeled off from the substrate and formed a floating conglomerate. Higher Ti-content in the film gave negative cytotoxicity. Morphology of cells incubated in extracts from films, which had in their composition less than 53% at. of Cu, was comparable to these from the control culture population (Figure 6).

The proliferation ability in indirect contact with thin films was determined with the aid of clonogenic test (Figure 7). The strong cytotoxicity of Cu film was confirmed. The 100% extract from this coating inhibited the L929 cell proliferation process. At higher dilution of the extract, the number of cells was above 70% of the control value. A similar toxic effect on cells was also found for the $Cu_{83}Ti_{17}$ coating. Reduction in the amount of copper in the film composition, down to 53 at.% even at the highest concentration (of the extract), did not inhibit proliferation (Figure 7). However, the most optimal environment for the development of the L929 is incubation with a $Cu_{25}Ti_{75}$ extract. Regardless, the extract concentration the number of cells was higher as-compared to the control sample or even to Ti film. This testifies the stimulating effect of $Cu_{25}Ti_{75}$ thin film on the proliferation of L929 cells.

In Figure 8, microscopic images of the degree of confluence and changes in the morphology of L929 cells in direct contact with thin films based on Cu and Ti (after 24 and 72 h of incubation) are shown. For Cu and $Cu_{83}Ti_{17}$ films, a zone of strong migration of copper ions from the sample can be observed, especially at the samples edge. Cells in this zone have been lysed and died immediately. More cells had correct morphology if they were placed further from the sample edge. Another example indicating on an unfavorable environment for the growth was a small confluence of cells (up to 30% after 24 h of incubation). In the case of $Cu_{53}Ti_{47}$, $Cu_{25}Ti_{75}$, and Ti films, morphological changes were significantly lower after 24 h of incubation. After 72 h, there was no significant difference in their confluence. This means that the amount of copper in the composition of thin films had a key influence on confluence and cell morphology.

| L929 cell culture with thin film extracts | | | | |
|---|---|---|---|---|
| **exposition time** | **24 hours** | | **72 hours** | |
| **concentration of the extract** | **100%** | **50%** | **100%** | **50%** |
| **Cu thin film** | | | | |
| **Cu$_{83}$Ti$_{17}$ thin film** | | | | |
| **Cu$_{53}$Ti$_{47}$ thin film** | | | | |
| **Cu$_{25}$Ti$_{75}$ thin film** | | | | |
| **Ti thin film** | | | | |

**Figure 6.** Microscopic photographs of the degree of confluence and changes in the morphology of L929 cells in contact with extracts (in 100% and 50% concentrations) of Cu, Cu-Ti, and Ti thin films after 24 and 72 h of incubation.

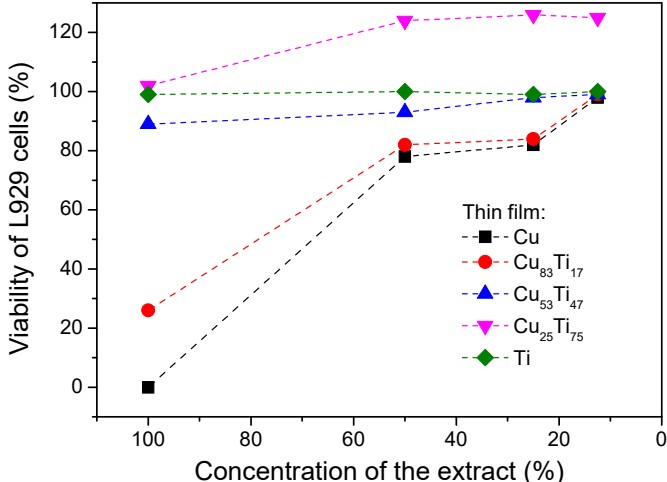

**Figure 7.** Results of the L929 fibroblast proliferation test in contact with various concentrations of extracts with Cu, Cu-Ti, and Ti thin films after 8 days of exposure.

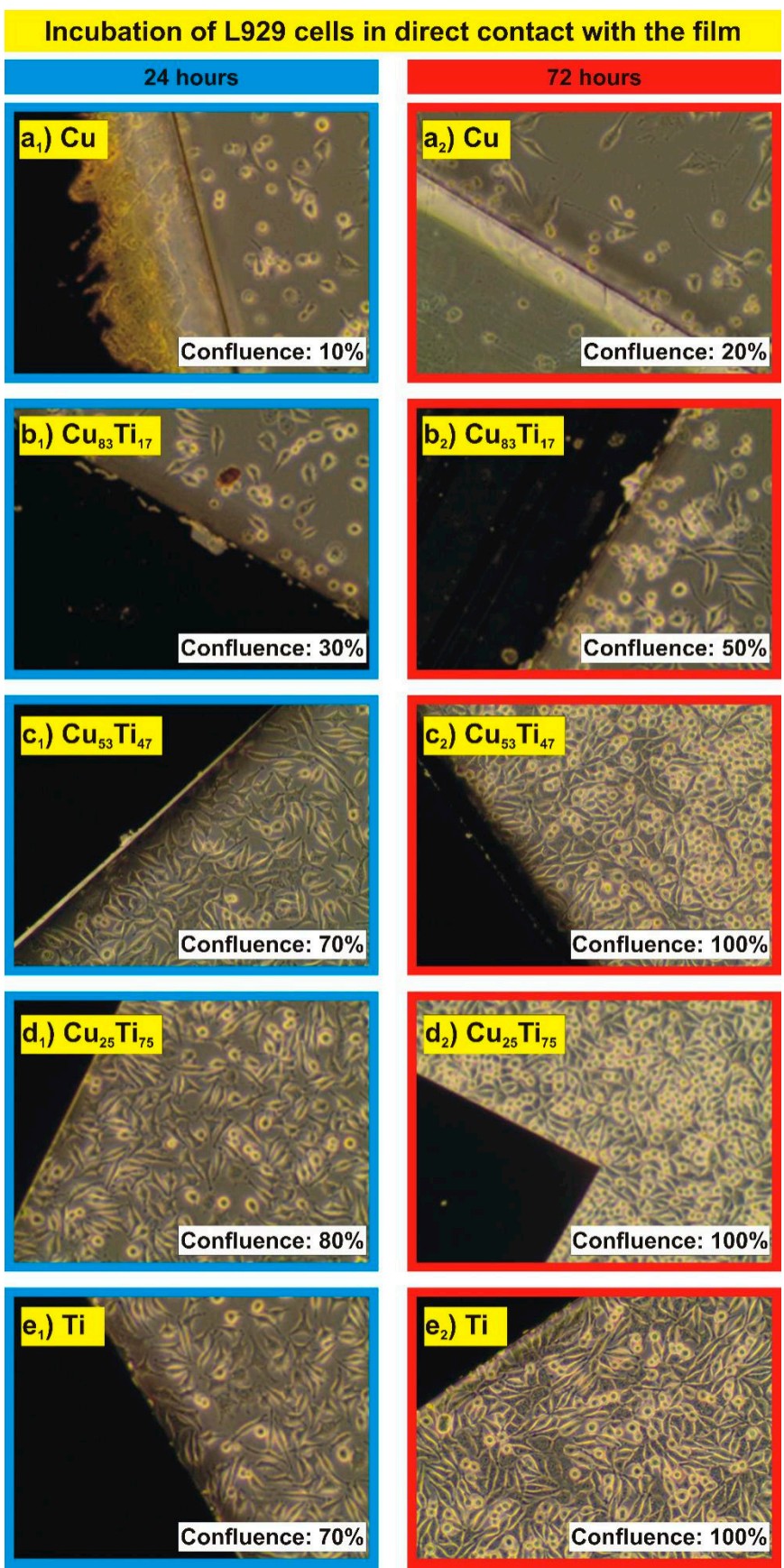

**Figure 8.** Microscopic photographs of the degree of confluence and changes in the morphology of L929 cells in direct contact with thin films based on Cu and Ti after 24 and 72 h of incubation.

## 4. Discussion

Research results allowed us to develop a comprehensive analysis of the interaction between prepared nanomaterials and eukaryotic cells (L929) and bacteria (*E. coli* and *S. aureus*). Figure 9 presents a graphical summary. Based on this model, it can be concluded that the increase of Cu content in the nanocrystalline Cu-Ti films results in the change of their structure. In the case of Cu thin film, the columnar character was obtained, while addition of titanium reduced the size of columns until a grainy structure could be obtained (for the Cu-Ti film with 25 at.% of Cu). Undoubtedly, this also determined the surface properties and bioactivity of the coatings.

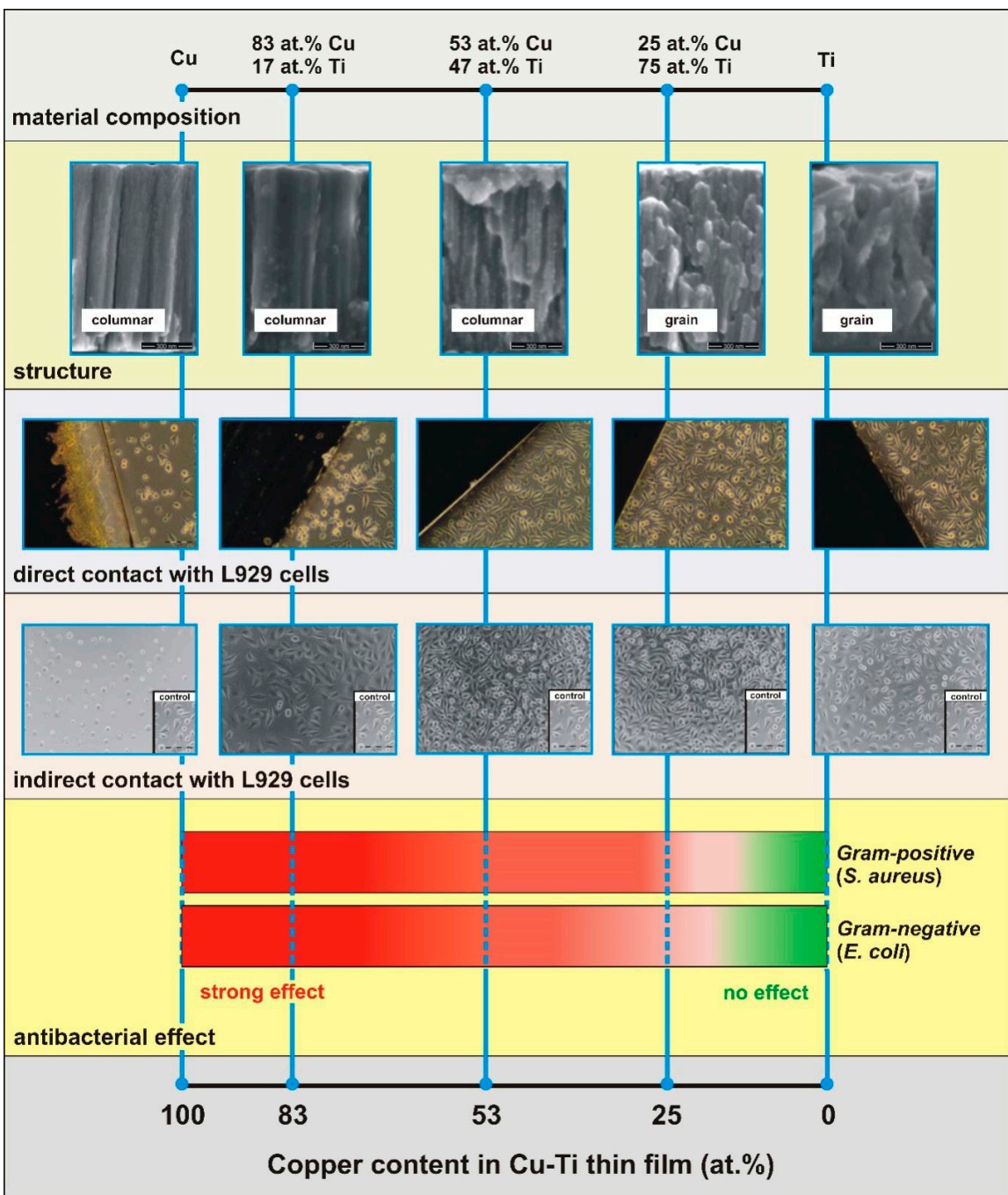

**Figure 9.** The influence of material composition on the biological properties of nanocrystalline thin films based on copper and titanium.

The amount of copper in Cu-Ti thin films had a significant impact on the physiology of mouse fibroblasts. Contact (direct and indirect) with $Cu_{83}Ti_{17}$ or Cu film was highly toxic and resulted in the death of fibroblasts. Less copper in the film than 53 at.% caused normal cell morphology after 24 h of direct contact, but, after 72 h, the number of cytoplasmic granules increased. Their presence may indicate on accumulation of copper ions in the cytoplasm. The different nature of cellular interactions with thin film was observed for $Cu_{25}Ti_{75}$. The confluence of L929 cells (after 24 h) for this coating was the highest in comparison with others that were made. This level was even higher than for fibroblasts exposure to the Ti film. Such an effects suggest stimulating properties of the nanocrystalline $Cu_{25}Ti_{75}$ coating. The diverse nature of interactions of Cu-Ti coatings indicates a different level of copper ion migration to the environment. The concentration of ions in the environment determines whether it has an inactive, destructive, or stimulating effect on various types of cells. Copper is an element necessary for the proper functioning of all cells [90]. For example, it should be delivered to the human body (an adult, weighing about 70 kg) at a dose of about 1.35 mg/day [91]. This means that its minimum amount to maintain balance in the body is around 18.86 ppb/day. Therefore, determining the amount of copper ions released from the nanocrystalline surfaces of Cu-Ti films to the environment was crucial to measurable assessment of their biological activity. The results of the AAS tests (Figure 10) showed that from the surface of Cu and Cu-Ti thin films migration of copper ions occurred, and this had an effect on live cells.

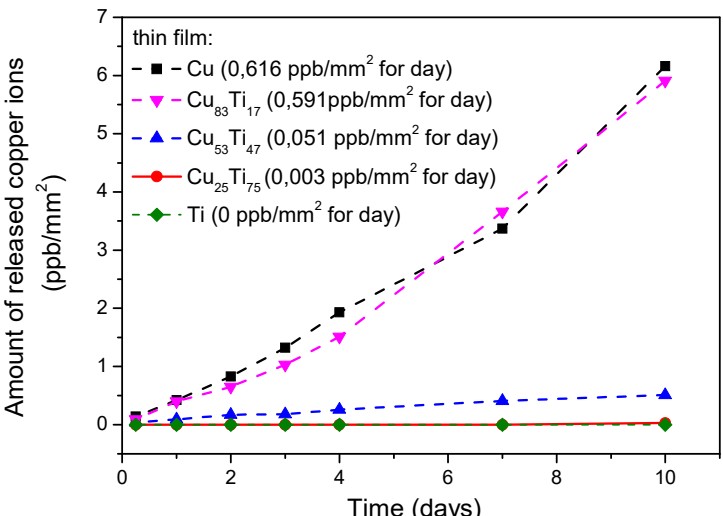

**Figure 10.** The amount of copper ions released into the environment from the surface of nanocrystalline thin films based on Cu and Ti determined by the AAS method.

In the case of the nanocrystalline $Cu_{83}Ti_{17}$ coating, the amount of released ions was close to the level of Cu film (0.591 ppb/mm$^2$ and 0.616 ppb/mm$^2$, respectively). Such a migration rate is sufficient to obtain a strong cytotoxic effect already in indirect contact with L929 fibroblasts. Reduction of copper content in Cu-Ti film down to 53 at.% resulted in the decrease of ion migration rate to 0.051 ppb/mm$^2$. This is approximately 8% of the level obtained for highly cytotoxic Cu and $Cu_{83}Ti_{17}$ films. Moreover, metabolic activity of L929 has not been disturbed in the indirect contact with $Cu_{53}Ti_{47}$ (lack of cytotoxicity). So, tenfold reduction of the migration rate, as-compared to the nanocrystalline Cu film, is sufficient to intermit toxic properties. In the case of Cu-Ti film with lower copper content, i.e., 25 at.%, the quantity of ions released into the environment per day was only 0.003 ppb/mm$^2$. This value is 0.5% of the amount of ions released from cytotoxic Cu and $Cu_{83}Ti_{17}$ coatings. This suggests that, with a 1:3 ratio of Cu:Ti content in the film, the self-passivation of titanium on the surface of the nanocrystalline coating is predominant and limits the migration of copper ions to the environment. This result is of particular interest because the percentage of L929 cells after 24 h of contact with the $Cu_{25}Ti_{75}$ extracts was even higher than the number of cells in the control culture. This indicates a

stimulating effect. Similar conclusions were drawn by Heidenau et al. [92]. They proved that a proper amount of copper ions gave the best effects in achieving a compromise between bactericidal and non-cytotoxic properties of L929 cells.

Except material composition and ion migration process, the degree of surface oxidation also had an effect on the bioactivity of the films. Analyzing the impact of Cu-Ti films on cell cultures, as well as on bacteria inactivation, it should be emphasized that the interaction may take a place either in direct contact—with the surface of the film—and in intermediate contact—with ions released to the surrounding environment. According to Park et al. [30] or Shedle et al. [58], $Cu^{1+}$ ions are more bioactive than $Cu^{2+}$. Such relation also occurs in contact with for *E. coli* and *S. aureus* [89]. Therefore, the percentage of $Cu^{0,1+}$ and $Cu^{2+}$ ions (based on the analysis of XPS spectra for the Cu2p state) was taken into account (Figure 11). In the case of films with a high copper content (Cu, $Cu_{83}Ti_{17}$, $Cu_{53}Ti_{47}$), the quantity of $Cu^{0,1+}$ ions was about two times higher compared to $Cu^{2+}$ ions. Nanocrystalline coating with smaller amount of copper (i.e., 25 at.%) was characterized by a nearly nine-fold more $Cu^{0,1+}$ ions than $Cu^{2+}$. This suggests that low amount of copper in the composition of the Cu-Ti thin film does not indicate on a low biological activity. These observations are also in agreement with other works [70].

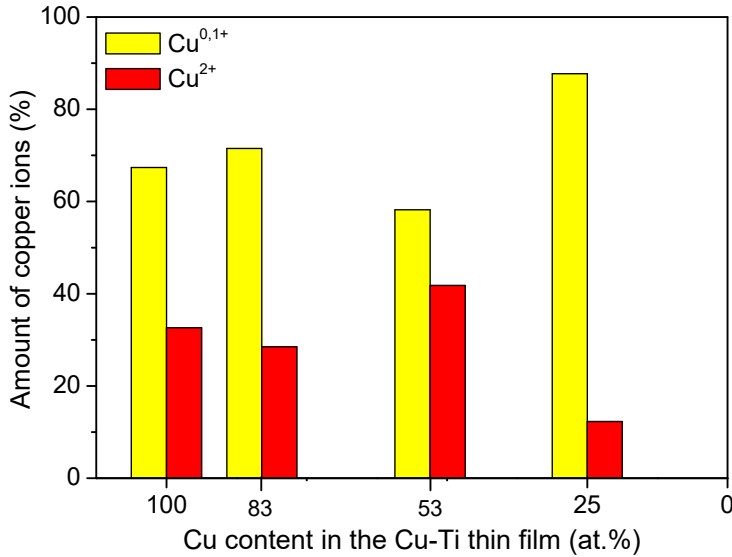

**Figure 11.** Influence of the Cu content in the nanocrystalline Cu-Ti thin film on the percentage amount of $Cu^{0,1+}$ and $Cu^{2+}$ ions on its surface.

Quantitative analysis revealed a number of dependencies between the material composition of thin films and their effect on eukaryotic cells and bacteria (Figure 12). In the case of films with a high copper content (Cu and $Cu_{83}Ti_{17}$) after 24 h of incubation, 100% of bacteria, both Gram-positive (*S. aureus*) and Gram-negative (*E. coli*), was killed. The same effect was obtained for mouse fibroblasts, although a small change in the film composition (addition of 17 at.% Ti) allowed us to maintain about 20% of the population. It should be added that a similar level of $Cu^{0,1+}$ and $Cu^{2+}$ ions was on the surface of both coatings. This means that the dynamics of interaction of these nanocrystalline films with L929 cells was determined by the migration process of Cu ions to the environment. In the case of analyzed coatings, the quantity of released ions was similar. Less copper in the Cu-Ti film, i.e., 53 at.%, allowed us to obtain a nanomaterial with strong antimicrobial properties (equal to Cu and $Cu_{83}Ti_{17}$ films), but the reduction of mitochondrial activity of L929 cells was not observed at the same time. Moreover, L929 viability was almost the same as for Ti film. It is worth adding that the percentage share of $Cu^{0,1+}$ ions in relation to $Cu^{2+}$ was 2:1. This means that the biological activity of as-deposited thin-film nanomaterials was determined by the amount of copper ions released from its surface. As

shown, the material with antibacterial and non-cytotoxic properties (Cu$_{53}$Ti$_{47}$ film) can be obtained due to limitation (10-times) of copper ion migration process from the surface.

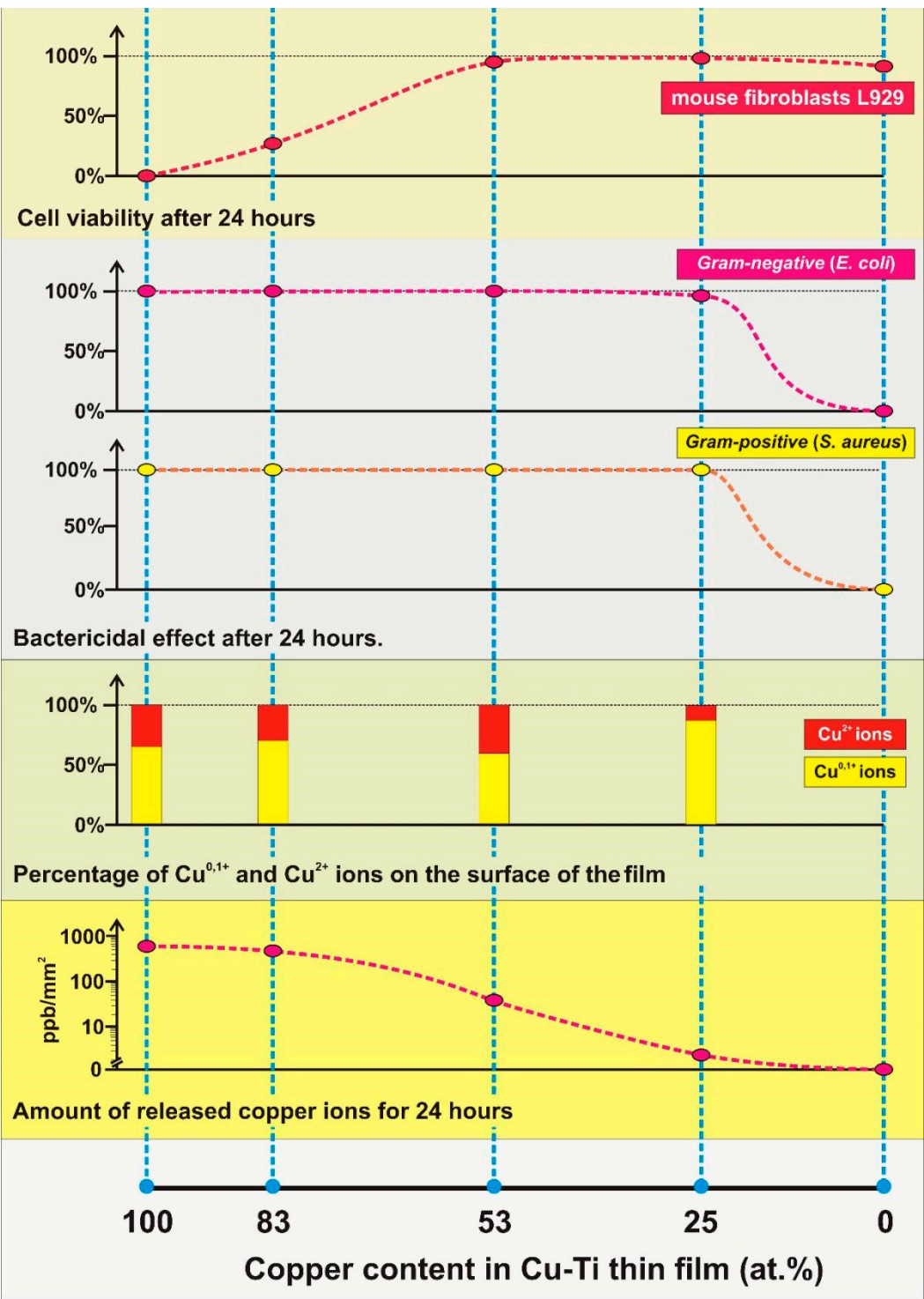

**Figure 12.** Quantitative results of the impact of material composition on biological properties (viability of L929 cells and bacteria *E. coli* and *S. aureus*), oxidation state, and amount of copper ions released from the surface of nanocrystalline Cu-Ti thin films.

Taking into account the future use of the studied thin-film nanomaterials, much more interesting results gave Cu$_{25}$Ti$_{75}$ film. The analysis of quantitative data revealed that this coating also exhibits a

high bactericidal level. There was a small difference (about 2%) in the percentage of killed bacteria that were representatives of different groups of microorganisms (Gram-positive and Gram-negative). However, the dynamics of bactericidal activity of both types of bacteria was similar. Convergent results were obtained by Shedle et al. [58] and Ma et al. [93]. Our results of MTT test have shown a larger number of metabolically active L929 cells as-compared to the biocompatible Ti film. This indicates on stimulation of their growth by $Cu_{25}Ti_{75}$ film. Heidenau et al. [92] and Zhang et al. [94] also pointed out the beneficial effect of copper on proliferation. They emphasized a stimulating effect of copper on the processes of angiogenesis and osteosynthesis. In bone marrow, stem cell viability studies concluded that the release of 0.16 ppb/day of $Cu^{2+}$ ions (to 0.9% NaCl) also resulted in a higher viability in contact with Cu-Ti material as-compared to Ti. Another indication of this relationship are studies of the impact of Cu-Ti surface on the level of cytotoxicity of MG-63 osteoblast. Lüthen et al. [95] noted that the amount of copper below 0.3 mmol/L stimulates cell proliferation, while the dose above 0.5 mmol/L was a limiting factor. Freedman et al. [96] determined the dose of copper which was not toxic to Morris hepatoma cells to be 200–400 pM. In contrast, for L929 fibroblasts in the study of materials intended for the production of intrauterine devices, the lethal dose was 46 μg/mL (~29 μM/24 h) [97].

It should be noted that the decrease of copper content in a coating down to 25 at.% limited the amount of released ions to 0.003 ppb/mm$^2$. This was too small to cause indirect cytotoxic effect on eukaryotic cells (for mouse fibroblasts L929) but sufficient to obtain antimicrobial properties (for *E. coli* and *S. aureus*). Finke et al. [73], in the studies of the influence of titanium materials with addition of copper on osteoblasts, concluded that the amount of copper ions in the range of 0.36 to 0.9 ppm is not cytotoxic to eukaryotic cells but enough to inhibit the growth of *S. aureus* bacteria. They concluded that titanium with the addition of copper damaged the peptidoglycan layer, leading to the separation of the cell membrane from the cell wall and increase its permeability. As a consequence, the cell content penetrates into the external environment, causing a cell shrink. The next stage of copper toxicity to bacteria was the effect on ROS overproduction, suppression of cellular respiration, and the effect on gene replication. Huang et al. [98] performed similar studies using different concentrations of Cu (from 0.01 to 10 mM). The results of these studies have shown that the most preferred concentration of Cu in the composition of Cu-Ti film was about 1 mM. This material was then anti-bacterial (for *S. aureus* and *E. coli*) and neutral for eukaryotic cells (mouse fibroblasts L929). It is worth adding that the method of exposure to the material (direct contact or indirect-on the extracts from the material) also has an impact on the level of biological activity. The studies of the influence of copper oxidation state (at the surface of the nanocrystalline films) suggest higher activity in direct contact. It was related to the fact that, on its surface, a much larger share (9:1) had $Cu^{0.1+}$ ions than $Cu^{2+}$.

## 5. Conclusions

In this work, the influence of material composition on structure and surface properties of bioactive coatings based on Cu and Ti was described. It was found that the level of bioactivity of Cu-Ti coatings can be modified by proper selection of copper content. All prepared films with copper were bactericidal (for *E. coli* and *S. aureus*). However, more important is the fact that the amount of copper had an impact on the viability of the L929 cell line. The cytotoxic, as well as stimulating, effect was observed in dependence on material composition of nanocrystalline Cu-Ti films. The bioactivity of the coatings was related to ion migration process and to the oxidation state of copper ions (amount of $Cu^{1+}$ ions). Such results allow preparation of innovative bioactive coatings with desired bio-impact on different types of microorganisms.

**Author Contributions:** Conceptualization, D.W., D.K. and B.S.; methodology, D.W., M.M., B.S., M.O. and A.O.; software, D.W. and M.M.; validation, D.W.; formal analysis, D.W., M.M., M.O. and B.S.; investigation, D.W., D.K., B.S., M.M., M.O., P.M. and A.O.; writing—original draft preparation, D.W., D.K., B.S., M.O. and M.M.; writing—review and editing, D.W.; visualization, D.W., M.M. and M.O.; supervision, D.W., D.K. and B.S.; project administration, D.W., D.K. and B.S.; funding acquisition, B.S. All authors have read and agreed to the published version of the manuscript.

**Funding:** This work was financed from the sources given by the Polish National Science Centre (NCN) as a research project number UMO-2016-21-B-ST8-02099.

**Conflicts of Interest:** The authors declare no conflict of interest. The funders had no role in the design of the study; in the collection, analyses, or interpretation of data; in the writing of the manuscript, or in the decision to publish the results.

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
