# Peer review of "Influence of Material Composition on Structure, Surface Properties and Biological Activity of Nanocrystalline Coatings Based on Cu and Ti"

_coatings, doi:10.3390/coatings10040343_

Round 1

Reviewer 1 Report

Dear Authors,

The present study deals with the evaluation of the influence of material composition on structure and surface properties of bioactive coatings based on Cu and Ti. Some specific changes in the structure and organization of the manuscript should be performed especially in the first part of the manuscript, as follows:

Introduction: This part should be re-organized. It should offer the context of the study more properly. Poor information on the Ti, compared with Cu, are given. Please add information on Ti, or shorten the Cu part, as it is not uniform. At the same time, please add information on the existing studies in scientific literature, that treat similar subject as your research subject and clearly state the originality and novelty of your study compared with them.

Materials and methods: Please separate methods from results. Some sections contain results, which should be found in the next section (e.g. film manufacturing). Please re-organize this section in a more logical order: preparation, physicochemical characterisation and biological testing.

            Moreover, authors should correct minor writing errors (e.g. name of bacterial species and strains should appear in italic, even in the abbreviations) and a language check should also be performed. If these changes are performed, the quality of the manuscript will surely increase.

Author Response

Answers to the report of Reviewer 1

on the manuscript entitled: Influence of material composition on structure, surface properties and biological activity of nanocrystalline coatings based on Cu and Ti

Authors: Damian Wojcieszak * , Malgorzata Osekowska , Danuta Kaczmarek , Bogumila Szponar , Michal Mazur , Piotr Mazur , Agata Obstarczyk

Authors :

We would like to express our gratitude for your remarks, which let us improve our manuscript. We have taken them into account in the revised version of my paper. Answering to the Reviewer’s remarks, We have introduced some revisions into the manuscript.

  1. Reviewer 1:

The present study deals with the evaluation of the influence of material composition on structure and surface properties of bioactive coatings based on Cu and Ti. Some specific changes in the structure and organization of the manuscript should be performed especially in the first part of the manuscript, as follows:

  1. a) Introduction: This part should be re-organized. It should offer the context of the study more properly. Poor information on the Ti, compared with Cu, are given. Please add information on Ti, or shorten the Cu part, as it is not uniform. At the same time, please add information on the existing studies in scientific literature, that treat similar subject as your research subject and clearly state the originality and novelty of your study compared with them.
  2. b) Materials and methods: Please separate methods from results. Some sections contain results, which should be found in the next section (e.g. film manufacturing). Please re-organize this section in a more logical order: preparation, physicochemical characterisation and biological testing.

1a. Authors :

According to the comment section Introduction was rewritten and extended. Information about properties of Ti was added. Moreover, more detailed description about properties of materials based on Cu-Ti and the novelty of our work was incorporated. All changes are marked in red.

1b. Authors :According to the comment section Materials and Methods was corrected. EDS results have been moved to the section Results. Moreover, the order of descriptions for individual research methods has been modified and ordered in accordance with the reviewer's comment.

  1. Reviewer 1:

Authors should correct minor writing errors (e.g. name of bacterial species and strains should appear in italic, even in the abbreviations) and a language check should also be performed. If these changes are performed, the quality of the manuscript will surely increase.

2. Authors :

According to the reviewer's comments, the text of the whole article has been checked and corrected. Changes in the text are marked in red.

Reviewer 2 Report

  1. The authors report novel type of coatings, however, more detailed comparison with the state of the art in the field is required. E.g. it is known that hydroxyapatite-coatings are state-of-the art in respect with the bioactive performance, e.g. DOI: 1016/j.msec.2018.12.045, DOI: 10.1016/j.matchar.2018.05.042.
  2. In fig. 1 it is better to show the whole range , since within a low energy range some traces of carbon can be seen, thus all elements concentration should be corrected, if required.
  3. It is known that bioactive performance of the coatings depend on the phase composition and the content of amorphous phase. Please provide some details on the bioactive performance of the field regarding to the degradation rate.
  4. In Fig. 7 standard deviations should be provided. Otherwise, no conclusions on the significance of the obtained values can be done. Please also provide standard deviations for all the graphs.

Author Response

Answers to the report of Reviewer 2

on the manuscript entitled: Influence of material composition on structure, surface properties and biological activity of nanocrystalline coatings based on Cu and Ti

Authors: Damian Wojcieszak * , Malgorzata Osekowska , Danuta Kaczmarek , Bogumila Szponar , Michal Mazur , Piotr Mazur , Agata Obstarczyk

Authors :

We would like to express our gratitude for your remarks, which let us improve our manuscript. We have taken them into account in the revised version of my paper. Answering to the Reviewer’s remarks, We have introduced some revisions into the manuscript.

  1. Reviewer 2:

The authors report novel type of coatings, however, more detailed comparison with the state of the art in the field is required. E.g. it is known that hydroxyapatite-coatings are state-of-the art in respect with the bioactive performance, e.g. DOI: 1016/j.msec.2018.12.045, DOI: 10.1016/j.matchar.2018.05.042.

1. Authors :

Authors agree with the reviewer's comments regarding the use of other materials in medicine than metals and their alloys. Therefore, relevant information has been incorporated into manuscript in the introduction section. The list of literature references was also extended. All changes in the text are marked in red.

  1. Reviewer 2:

In fig. 1 it is better to show the whole range since within a low energy range some traces of carbon can be seen, thus all elements concentration should be corrected, if required.

2. Authors :

The authors agree with the reviewer's opinion that presentation the low energy range on a EDS scale would provide additional information about the carbon. However, our goal was to show the range, which was important to determine the percentage of titanium and copper. Film composition was determined based on the analysis of CuLa and TiKb lines, which are in the range of 3 eV ÷ 9 eV. Carbon, as the reviewer rightly noted, is present in the composition of our films due to, for example exposition to the ambient air for several days. Authors would also like to mention that the presence of carbon was used in the analysis of XPS spectra for calibration. The intensity and shape of all XPS spectra for the C1s state were similar and relatively low, hence it was not taken for analysis.

  1. Reviewer 2:

It is known that bioactive performance of the coatings depend on the phase composition and the content of amorphous phase. Please provide some details on the bioactive performance of the field regarding to the degradation rate.

3. Authors :We agree with the reviewer's comment on the impact of the amorphous phase on the properties of materials, including their biological activity. We have observed such effects in the case of oxide materials. In the case of discussed article only metallic films were the subject of the study. In their case presence of amorphous phase was not identified. Stability of Cu-Ti films was the subject of previous work [Wojcieszak, D.; Kaczmarek, D.; Antosiak, A.; Mazur, M.; Rybak, Z.; Rusak, A.; Osekowska, M.; Poniedzialek, A.; Gamian, A.; Szponar, B. Influence of Cu-Ti thin film surface properties on antimicrobial activity and viability of living cells. Materials Science and Engineering: C 2015, 56, pp. 48-56.]. We have found that metallic copper coating in an aqueous environment or in a contact with agar (nutrient medium for bacteria) was dissolved – a strong effect of ion migration to the environment was observed. Our research has shown that 24 hours is enough for complete degradation of a film with a thickness of 100 nm (especially at the edges of the films). In contrast, titanium had very high corrosion resistance. Hence the results of our research showed that already 10 at. % of Ti additive can significantly improve corrosion resistance of copper.  

  1. Reviewer 2:

In Fig. 7 standard deviations should be provided. Otherwise, no conclusions on the significance of the obtained values can be done. Please also provide standard deviations for all the graphs.

4. Authors :

Authors agree with the reviewer's comment regarding the necessity to provide the accuracy with which the test results were estimated. Due to the complexity of the charts with various measurement results, relevant information was provided in the Materials and Methods section. All changes in the text are marked in red.

Round 2

Reviewer 1 Report

Dear Authors,

After performing the suggested changes, quality of the manuscript surely increased. A minor change should nevertheless be performed in the abstract: please change the 2 sentences that begin in the same way (lines 22 and 25).

Reviewer 2 Report

The manuscript was improved and can now be accepted for publication.